# Modulation of Hemostasis in COVID-19; Blood Platelets May Be Important Pieces in the COVID-19 Puzzle

**DOI:** 10.3390/pathogens10030370

**Published:** 2021-03-19

**Authors:** Magdalena Ulanowska, Beata Olas

**Affiliations:** Department of General Biochemistry, University of Lodz, Pomorska 141/3, 90-236 Lodz, Poland; magdlena.ulanowska@edu.uni.lodz.pl

**Keywords:** COVID-19, cardiovascular disease, hemostasis

## Abstract

Although the precise pathogenesis of coronavirus disease 2019 (COVID-19) currently remains unknown, its complex nature is gradually being revealed. COVID-19 is a disease caused by the SARS-CoV-2 virus and leads to respiratory dysfunction. Studies on hemostatic parameters have showed that COVID-19 significantly affects the disruption of the coagulation system and may contribute to coagulation and thrombotic events. A relevant cause of hemostasis disorders is inflammation and cytokine storms, which cause, for example, endothelial dysfunction in blood vessels. In order to prevent and treat states of hypercoagulability and thrombosis, the administration of anticoagulants, e.g., heparin, is recommended. The present mini-review describes the relationship between hemostasis and COVID-19, and discusses whether this relationship may cast light on the nature of COVID-19. The present short manuscript also examines the relationship between blood platelets and COVID-19. In addition, the paper explores the potential use of antiplatelet drugs in COVID-19 cases. The studies were identified by searching electronic databases, including PubMed and SCOPUS.

## 1. Introduction

Hemostasis is a set of mechanisms aimed at, on the one hand, ensuring the fluidity of blood in the blood vessels (preventing excessive clotting) and, on the other hand, the rapid inhibition of bleeding when the vessel wall is interrupted. Hemostasis ensures a balance between procoagulation and anticoagulant mechanisms; many elements are involved in this process—platelets, blood vessel walls with endothelium, the coagulation systems (including coagulation factors) and fibrinolysis (plasmin) [1,2]. In many patients with COVID-19, hemostasis disorders have been observed, which increase the risk of developing DIC (disseminated intravascular coagulation) and coagulopathy [3]. DIC is a serious disease associated with the overstimulation of the coagulation system, which leads to the formation of microvascular thrombosis, which is associated with an increased risk of bleeding with hyperfibrinolysis and even organ failure.

Coronavirus disease 2019 (COVID-19) is a disease caused by the SARS-CoV-2 virus —a new strain of RNA viruses, one of the beta-coronaviruses. COVID-19 is referred to as severe acute respiratory distress syndrome, and this virus contributes to respiratory failure and ARDS (acute respiratory distress syndrome)—A disorder of pulmonary inflammation, which leads to acute hypoxemic respiratory failure. Preliminary results indicate that patients affected by COVID-19 also demonstrate various hemostasis dysfunctions, such as coagulation dysfunction, this being a major cause of death [4]. Moreover, the risk of thrombotic events can also be increased by the activity of the inflammatory response, the clinical illness, and by various traditional risk factors. Adverse drug–drug interactions may also occur between COVID-19 treatments and antiplatelet agents. Controlling hemostasis parameters can help to establish predictors of poor prognosis in infected patients [3,5,6]. Therefore, it is important to understand the mechanisms of the virus’ action on the coagulation system and to develop an effective treatment preventing the dangerous consequences of the disease [7]. The present mini-review describes the relationship between hemostasis and COVID-19, and discusses whether this relationship may cast light on the nature of COVID-19. In addition, the participation of blood platelets in the pathogenesis of COVID-19 remains poorly understood, particularly the influence of the mean size and reticulation, and the stages of platelet activation, such as aggregation. This short manuscript also describes the role of blood platelets in COVID-19. It discusses the potential use of antiplatelet medication in treating COVID-19 based on a review of studies identified in electronic databases, including PubMed and SCOPUS. The last search was run on 15 December 2020.

## 2. Abnormalities in the Parameters of Hemostasis and COVID-19

A high number of thrombotic episodes have been described in histopathology reports and clinical cases of COVID-19, and it has been suggested that infection may constitute an important risk factor for thrombosis [8]: both venous thrombosis, i.e., venous sinus thrombosis and deep vein thrombosis, and arterial thrombosis, i.e., myocardial infarction and stroke. Lax et al. [9]. report the occurrence of pulmonary arterial thrombosis in COVID-19 patients. In cases of pulmonary microthrombosis, the damage to endothelial cells results in the overactivation, aggregation and retention of blood platelets, and the formation of a thrombus at the injured side; this may lead to the depletion of platelets and megakaryocytes, resulting in the decreased production and increased consumption of blood platelets [10].

Abnormalities in coagulatory parameters have been observed in many COVID-19 patients, including increased levels of D-dimer, increased concentrations of fibrinogen and prolonged prothrombin times (PTs), thrombotin times (TTs), and activated partial thromboplastin times (APTTs) [7,11,12]. These changes can predict the severity and prognosis of COVID-19 [7].

It is very likely that COVID-19 is associated with a hypercoagulable state, and the clinical features of COVID-19-associated coagulopathy differ from DIC—DIC only occurs in the late stage of the disease in severe cases. COVID-19-related coagulopathy is associated with a high risk of morbidity and mortality, is multifactorial in nature, and significantly increases the likelihood of death [11,12]. One of the causes of hemostasis disorders is the occurrence of severe inflammation and cytokine storms that cause the endothelial dysfunction of the blood vessels and promote the spread of thrombosis [8,12,13]. As a result, pro-inflammatory cytokines can significantly affect damage to the vascular endothelium, abnormal clot formation, and the excessive activation of the coagulation system and blood platelets, and inhibit the fibrinolytic and anticoagulant system. It causes clotting that can lead to the development of DIC and microcirculation disorders, and finally to severe multi-organ dysfunction [7,14]. For people with COVID-19, significantly increased levels of IL-6 were observed in patients who died compared with those in survivors. In hospitalized patients with respiratory failure, IL-6 is an important parameter of disease progression. In addition, C-reactive protein (CRP), an acute-phase protein produced in the liver, is a vital marker for people with COVID-19 because its rapid increase is observed in most severe cases [15]. A study by Tang et al. [5] on a group of patients confirmed that patients with COVID-19 who died had significantly higher levels of D-dimers and FDP (fibrinogen and fibrin degradation products), and longer PTs and APTTs compared with the survivors. Another parameter of hemostasis to be investigated is increased thrombin time [16].

There are many risk factors for thrombosis in patients with COVID-19 (Figure 1), including old age, male sex, long-term immobilization and the occurrence of cardiovascular diseases, especially hypertension [14].

The International Society of Thrombosis and Haemostasis (ISTH) recommends the routine testing and control of hemostasis parameters in all hospitalized patients. Coagulopathies associated with COVID-19 infection are correlated with a high incidence of thrombotic events, while bleeding complications are very rare; therefore, standard anticoagulation is recommended [13]. Abnormalities in hemostasis parameters in patients with COVID-19 are demonstrated in Figure 2.

Iba et al. [13]. described significant differences and similarities regarding parameters, including the platelet count, D-dimer and fibrinogen concentrations, inflammatory cytokine levels and others, in patients with COVID-19 and diseases related to blood clotting disorders (e.g., SIC and DIC), which are presented in Table 1. This comparison may be helpful in identifying bleeding disorders and choosing appropriate treatments for the particular diseases.

## 3. Levels of D-Dimer and Fibrinogen in COVID-19

D-dimer is a product of the degradation of crosslinked (stabilized) fibrin, and an elevated level of this parameter suggests the occurrence of thrombosis and even pulmonary embolism in severe cases of COVID-19, which is associated with increased fibrinolytic activity in the organism [7,14]. Additionally, fibrinogen is an important indicator of the hypercoagulable state, high levels of which are observed in COVID-19 patients [17]. Fibrinogen is an acute-phase protein produced in the liver that plays an important role in the organism’s defense against invading pathogens, as it regulates the antimicrobial activity of immune cells and is involved in the formation of clots, thereby reducing the spread of the pathogen. Bi et al. [18] showed significant differences in clotting parameters in people with COVID-19. Severe cases had higher levels of fibrinogen and D-dimer at the earliest stages, and their levels fluctuated as the disease progressed. It is presumed that COVID-19 patients show increased coagulation and fibrinolysis activity, characterized by, among other things, an increased concentration of D-dimers. However, measuring fibrinogen in non-survivors showed a marked drop in fibrinogen levels shortly before death [4].

The reason for the changes observed in the measurement of these parameters was described by Thachil [19]. Patients with COVID-19 initially have elevated levels of fibrinogen as a result of the body’s developing inflammatory response and immune defense. At the same time, there is a slight increase in the level of D-dimers. However, during long-term inflammation, platelet granules cease to be released and fibrinogen levels drop significantly, while D-dimer levels rise rapidly. In this situation, clots are formed.

## 4. Blood Platelet Count and Platelet Activation in COVID-19

Hematological dysfunction has also been described in COVID-19 patients [20,21]. For example, Lu and Wang [22] report dynamic changes in routine blood parameters, including blood platelets, as well as a relationship between thrombocytopenia and COVID-19. In healthy adult individuals, the normal number of blood platelets is between 1.5 × 10^5^ and 4.5 × 10^5^ platelets per microliter of blood (150–450 × 10^9^/L). Thrombocytopenia is a disease when the platelet count is below the reference value [23]. The main function of blood platelets in hemostasis is to stop bleeding in the event of damage to blood vessels. Thrombocytopenia can be caused by a genetic condition or an autoimmune response and leads to coagulation disorders—coagulopathy. Thrombocytopenia can also result from the overactivation and wear of platelets associated with thrombosis [24]. A meta-analysis by Violi et al. [25] showed that the majority of severe cases of COVID-19 show significant decreases in the platelet count—down to around 1.0 × 10^5^ µL and, in some cases, below this value. Bao et al. [26] reported in their study that the platelet count was much lower in severe cases of COVID-19—about 186.00 (103.50–249.00) × 10^9^/L—than non-severe patients—where it was about 251.00 (202.00–317.00) × 10^9^/L. A meta-analysis of nine experiments, including a total of 1799 patients with COVID-19, also indicated that blood platelet counts were significantly lower in patients with more severe COVID-19; however, the heterogeneity of the experiments was high [27]. In addition, abnormalities in blood platelet counts are relatively uncommon in initial presentations of COVID-19 [28]. It has been found that 5 to 18% of the entire population demonstrated a low blood platelet count of around 100,000 µL or lower [25]. Furthermore, Bao et al. [26] report that six out of seven COVID-19 patients who died suffered thrombocytopenia during hospitalization, and that the blood platelet count decreased subsequently until death. Therefore, it is unclear whether thrombocytopenia can be used as a clinical biomarker for COVID-19. However, Zhao et al. [29] recommends including blood platelet count as an additional parameter in the current management process for COVID-19 patients. Other authors [30,31] suggest that thrombocytopenia, generally moderate, may be a marker of the severity of COVID-19.

Amgalan and Othman [20] propose several possible mechanisms for the induction of thrombocytopenia by COVID-19 (Figure 3). Infection may induce a cytokine storm, or the virus may infect bone marrow cells via CD13/CD66a, destroying the cells and inhibiting hematopoiesis [31]. Alternatively, COVID-19 may induce the production of autoantibodies or immune complexes that target blood platelets, resulting in thrombocytopenia [20].

Interestingly, it has recently been reported that platelet counts may be increased in COVID-19 patients, and three possible mechanisms have been proposed [21,30,32]. The first possibility is that infection induces a cytokine storm (including thrombopoietin and interleukin-3, -6, -9, and -11) stimulating megakaryopoiesis and/or thrombopoiesis. The second is that endothelial injury may occur, leading to the release of von Willebrand factor (vWF), which may interact with megakaryocytes via GPIb/vWF. Finally, it is possible that thrombopoietin is released, stimulating the activity of megakaryocytes. In addition, however, megakaryocytes have been observed to produce blood platelets within the alveolar capillaries of COVID-19 patients [33]. In addition to altered blood platelet numbers, COVID-19 patients also present with alterations in a number of other elements of hemostasis, including coagulation times (thrombin times and prothrombin times), d-dimer concentrations and fibrinogen levels [4,25,34,35].

As blood platelets clearly play an important role in hemostasis, immune defense and the inflammation process, platelet counts may provide valuable data in the treatment of COVID-19. In addition, COVID-19 patients also demonstrate altered modulation of platelet activation. Future studies based on sophisticated methodology should screen for blood platelet activation in cases of COVID-19, particularly regarding the action of blood platelet activation factors such as GPIIb/IIIa, P-selectin and TXA_2_. The results concerning the role of blood platelets may provide crucial information supporting the prophylaxis and treatment of COVID-19.

Due to platelets playing a very important role in hemostasis, and inflammatory and immune responses, it is suggested that the control of changes in the platelet count may be a very important prognostic parameter in the assessment of disease severity and prognosis in COVID-19 [20].

Patients with thrombocytopenia are at risk of bleeding, and most guidelines recommend platelet transfusions when their platelet counts are in the range of 30–50 × 10^9^/L, when bleeding occurs or in people at high risk of bleeding [36].

Blood platelet activation, indicated by the P-selectin or thromboxane A_2_ (TXA_2_) level, is believed to be influenced by changes in vascular and coagulation profiles, as are reactive oxygen species (ROS) production and NADPH oxidase (NOX2) activation. However, it remains unknown whether these factors are altered in COVID-19 patients. In addition, many case studies have examined patients demonstrating traditional risk factors for thrombosis, including obesity and diabetes [8]. Baergen et al. [37] suggest that pregnancy may increase the risk of thrombosis in women with COVID-19, and Cui et al. [38] indicate that approximately 25% of patients with severe COVID-19 may be at high risk of venous thromboembolism.

It has been hypothesized that, during COVID-19 infection, platelets are stimulated and recruited at the site of infection, including the lungs. These participate in the activation of the inflammatory process, as well as in the appearance of complications related to coagulopathy. Moreover, larger platelets contain higher numbers of dense granules and produce more active compounds, including TXA_2_, and are believed to be consequently more reactive than smaller platelets, with greater prothrombotic activity. Manne et al. [39] indicate that the observed increase in blood platelet activation may also partially be attributed to increased mitogen-activated protein kinase (MAPK) pathway activation.

Blood platelet activation and aggregation are associated with various receptors. Most platelet agonists activate blood platelets via G protein-coupled receptors, which may also act as targets for different antiplatelet drugs. It has been proposed that, in cases of COVID-19, blood platelet activation may be dysregulated via alterations in the function of angiotensin converting enzyme (ACE), and by changes in the angiotensin II/angiotensin type 1 receptor: a G-protein coupled receptor [8,40]. Another key mediator of platelet activation is proteinase-activated receptor 1 (PAR1), which is, in turn, mediated by thrombin, a serine protease. Some selective PAR1 antagonists have been used for the treatment of CVDs. One of these is the platelet activation inhibitor vorapaxar, which has been approved for treating myocardial infarction. It is possible that PAR1 may be a therapeutic target in COVID-19; however, before clinical trials can be carried out, the further preclinical validation of PAR1 antagonists such as vorapaxar is needed, as vorapaxar use has been associated with an increased risk of bleeding [41]. Preclinical studies underway with animal models should clarify the mechanisms behind the potential beneficial effects of PAR1 inhibition in vivo, including the actions on blood platelets [42].

In addition to its anti-inflammatory, antiplatelet and antithrombotic properties, aspirin has also demonstrated antiviral activity against DNA and RNA viruses, including various human coronaviruses. Bianconi et al. [43] suggest that its antiviral actions may be realized through three main pathways, cyclo-oxygenase-2, nuclear factor kappa beta and heme-oxygenase-1, all of which are modulated by aspirin. However, it remains unclear whether aspirin may be a safe and effective therapeutic candidate in COVID-19 patients, as clinical studies investigating the effects of different doses of aspirin in COVID-19 patients are often inconclusive. Moreover, such antiplatelet drugs can potentiate the action of anticoagulation. For example, in a study of COVID-19 patients (*n* = 5) with pulmonary infiltrates and D-dimer levels more than three times above the normal upper limit, those who received aspirin and clopidogrel demonstrated a higher PaO_2_/FiO_2_ ratio after 48 h than controls; in addition, these levels persisted for seven days [44].

## 5. Treatment and Prevention

Despite the introduction of vaccines against COVID-19, the pandemic continues, and no effective drug has been developed to combat the infection and its dangerous consequences, often related to the clotting system.

Since COVID-19 is associated with the occurrence of hypercoagulability, the careful observation and evaluation of laboratory parameters related to hemostasis at the start of treatment and throughout the course of the disease is very important in developing an individualized approach to the treatment or prevention of thrombotic events. Antithrombotic prevention and the early identification of the possible consequences of infection, e.g., coagulopathy, DIC, and venous thromboembolism, are likely to reduce mortality in infected patients and help to prevent dangerous complications. It is necessary to develop an anticoagulant treatment regimen using, for example, pharmacology (antithrombin or thrombomodulin) in patients with COVID-19 having abnormalities in coagulation parameters [45].

In order to prevent the occurrence of hypercoagulability, the use of low-molecular-weight heparin with proven anticoagulant and anti-inflammatory properties is also suggested. Patients with COVID-19 should be on thromboprophylaxis with the option of full therapeutic anticoagulation or tissue plasminogen activator in high-risk patients. Both The International Society on Thrombosis and Hemostasis (ISTH) and the American College of Cardiology recommend that patients with COVID-19 with respiratory failure and comorbidities be given low-molecular-weight heparin (LMWH) as thromboprophylaxis [46,47]. Tang et al. [48] confirmed the properties of heparin in a study on a group of patients with COVID-19. The use of LMWH contributed to a decrease in mortality and improved parameters in patients with significantly elevated levels of D-dimer and fibrinogen. Though the implementation of antithrombotic prophylaxis, there are still many cases of thromboembolism in hospitalized COVID-19 patients, so it is important to conduct randomized clinical trials to develop effective and safe prevention and treatment [46]. Although the use of heparin in patients with COVID-19 is recommended, no specific dose has been established that is effective and safe for each patient. It should be remembered that increased doses of anticoagulants may influence the risk of more frequent bleeding episodes. In this situation, the best solution seems to be to conduct randomized trials [14,49].

Recently, there have been studies involving combination therapy with nafamostat and heparin. Nafamostat is a serine proteinase inhibitor used, among other factors, in treating disseminated intravascular coagulation (DIC) [50]. A study by Takahashi et al. [51] was conducted in a patient with COVID-19 pneumonia, hypoxemia and pulmonary embolism. The use of a combination of heparin and nafamostat resulted in significant improvements, which is promising for the effective treatment of severely ill COVID-19 patients. The advantage of nafamostat is that it does not cause bleeding as a side effect—unlike heparin, it also has antiviral effects and could be effective in treating DIC in patients with COVID-19. Unfortunately, it has low anticoagulant activity [36]. In addition, drug–drug interaction may occur between antiplatelet compounds and anticoagulants in COVID-19 cases, resulting in an increased risk of thrombosis. Possible treatment strategies are listed in Figure 4.

## Figures and Tables

**Figure 1 pathogens-10-00370-f001:**
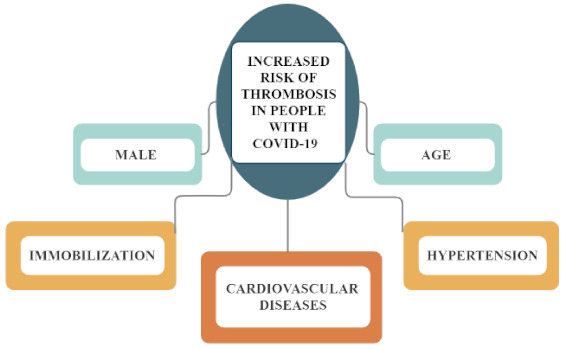
Factors that increase the risk of thrombosis in people with COVID-19.

**Figure 2 pathogens-10-00370-f002:**
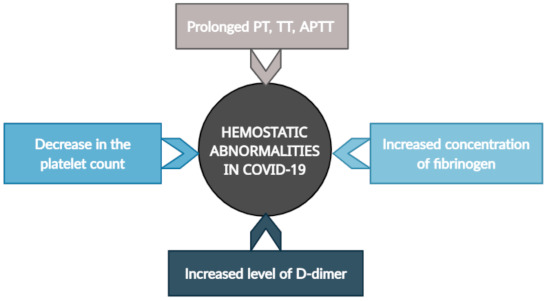
Abnormalities in hemostasis parameters in patients with COVID-19. PT—prothrombin time, TT—thrombin time, APTT—activated partial thromboplastin time.

**Figure 3 pathogens-10-00370-f003:**
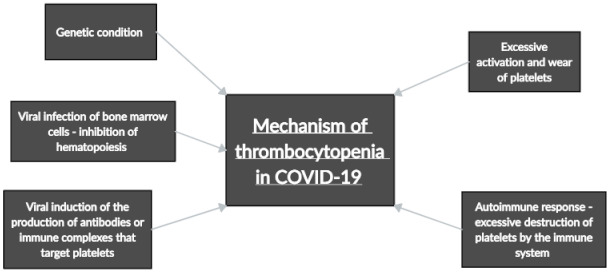
Possible mechanism of thrombocytopenia in COVID-19 patients.

**Figure 4 pathogens-10-00370-f004:**
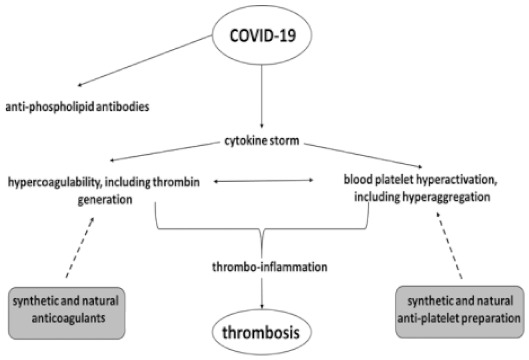
Possible mechanism for the increased thrombotic risk in COVID-19 patients and possible treatment strategies [4,43], modified.

**Table 1 pathogens-10-00370-t001:** The similarities and differences in thrombosis according to laboratory data in patients with COVID-19 and coagulation diseases [13], modified.

	Primary Cause and Target of Coagulopathy	Thromboembolism	Platelet Count	D-Dimer	PT/aPTT	Fibrinogen	Antithrombin	Activated Complement System	Inflammatory Cytokines (IL-1β, IL-6)	Antiphospholipid Antibody
**COVID-19**	Macrophage/endothelial cell	Microthrombosis/venous thrombosis	↑~↓	↑	→~↑	↑	→	+	↑	+
**DIC/SIC**	Macrophage/endothelial cell	Microthrombosis	↓	↑	↑	→~↓	↓	−	↑	−
**HPS**	Inflammatory cytokines	Microthrombosis/venous thrombosis	↓	→	→	→	→	−	↑	−
**APS**	Antiphospholipid antibody	Arterial/venous thrombosis	↓	→	PT → aPTT↑	→	→	−	−	+
**TMA (aHUS/TTP)**	Complement system/ADAMTS13	Microthrombosis or arterial/venous thrombosis	↓	→~↓	→	→	→	aHUS +/− TTP −/+	−	−

ADAMTS13—a disintegrin and metalloproteinase with a thrombospondin type 1 motif, member 13; aHUS—atypical hemolytic uremic syndrome; APS—antiphospholipid syndrome; aPTT—activated partial thromboplastin time; DIC—disseminated intravascular coagulation; HPS—hemophagocytic syndrome; IL—interleukin; SIC—sepsis-induced coagulopathy; PT—prothrombin time; TMA—thrombotic microangiopathy; TTP—thrombotic thrombocytopenic purpura; VWF—von Willebrand factor. ↑ increase; ↓ decrease; → no changes; + changes; − no changes

## Data Availability

This is a review paper, this item does not apply to this review.

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
