# Peer review of "Modulation of Hemostasis in COVID-19; Blood Platelets May Be Important Pieces in the COVID-19 Puzzle"

_pathogens, 2021, doi:10.3390/pathogens10030370_

Round 1

Reviewer 1 Report

The review is very interesting and of great interest. Overall, the review is generally well written however it does lack details at times. This makes it not easy for the reader if they do not have a background in the discussed disorders. Some examples are:

First paragraph - DIC is not explained

“Abnormalities in coagulatory parameters have been observed in many COVID-19

patients, including D-dimer levels, fibrinogen concentration, prothrombin time (PT), thrombotin time (TT), and activated partial thromboplastin time (APTT).11,7,12 These

changes can predict the severity and prognosis of COVID-19.” There is no discussion as to what these parameters are and their context.

Figure 4 – could the markers discussed throughout be incorporated to give readers a clearer understanding of the processes and how they present in covid?

Author Response

The review is very interesting and of great interest. Overall, the review is generally well written however it does lack details at times. This makes it not easy for the reader if they do not have a background in the discussed disorders. Some examples are:

First paragraph - DIC is not explained

Response: We have added definitions to DIC and ARDS on page 1. 

„DIC is a serious disease associated with overstimulation of the coagulation system, which leads to the formation of microvascular thrombosis, what is associated with an increased risk of bleeding with hyperfibrinolysis and even organ failure.”

“ARDS (Acute Respiratory Distress Syndrome) - disorder of pulmonary inflammation, which leads to acute hypoxemic respiratory failure.”

Abnormalities in coagulatory parameters have been observed in many COVID-19 patients, including D-dimer levels, fibrinogen concentration, prothrombin time (PT), thrombotin time (TT), and activated partial thromboplastin time (APTT).11,7,12 These changes can predict the severity and prognosis of COVID-19.” There is no discussion as to what these parameters are and their context.

Response: We have characterized changes in hemostatic parameters with COVID-19 on page 2.

“Abnormalities in coagulatory parameters have been observed in many COVID-19 patients, including increased level of D-dimer, increased concentration of fibrinogen and prolonged prothrombin time (PT), thrombotin time (TT), and activated partial thromboplastin time (APTT).”

Figure 4 – could the markers discussed throughout be incorporated to give readers a clearer understanding of the processes and how they present in covid?

Response: The chapter “Treatment and prevention” describes Figure 4.

Reviewer 2 Report

In this review, the authors sought to describe the relationship between hemostasis and COVID-19. 

  • The English language need extensive editing. A large portion of the manuscript is not readable!
  • The authors are encouraged to rewrite the abstract. It isn't a concise abstract in current shape.
  • The main text need reorganization. There is no logical structural subtitles, therefore readers may be easily lost.
  • All Figures look like drafts. I suggested polishing figures.
  • Figure 3 provides little useful information regarding the mechanism of thrombocytopenia. The authors should put more effort on how COVID-19 results in 3 possible ways of platelet decrease.  

Author Response

In this review, the authors sought to describe the relationship between hemostasis and COVID-19. 

  • The English language need extensive editing. A large portion of the manuscript is not readable!

Response: Our manuscript was corrected by a native speaker of English (firm – Adhoc English).

  • The authors are encouraged to rewrite the abstract. It isn't a concise abstract in current shape.

Response: Abstract has been shortened, now fits into the required length.

  • The main text need reorganization. There is no logical structural subtitles, therefore readers may be easily lost.

Response: We have changed the subtitle on page 2.

Previously: Hemostasis disorders and the COVID-19

Now: Abnormalities in the parameters of hemostasis and the COVID-19

  • All Figures look like drafts. I suggested polishing figures.

Response: Figures 1, 2 and 3 have been completely revised.

  • Figure 3 provides little useful information regarding the mechanism of thrombocytopenia. The authors should put more effort on how COVID-19 results in 3 possible ways of platelet decrease.  

Response: Now, Figure 3 provides more detail on the mechanism of thrombocytopenia.

Round 2

Reviewer 2 Report

In the revised manuscript, the authors have addressed my concerns. The manuscript is improved substantially.